# The Use of Tissue-on-Chip Technology to Focus the Search for Extracellular Vesicle miRNA Biomarkers in Thyroid Disease

**DOI:** 10.3390/ijms25010071

**Published:** 2023-12-20

**Authors:** Thomas Haigh, Hannah Beattie, Mark A. Wade, James England, Dmitriy Kuvshinov, Laszlo Karsai, John Greenman, Victoria Green

**Affiliations:** 1Centre for Biomedicine, Faculty of Health Sciences, Hull York Medical School, University of Hull, Hull HU6 7RX, UK; t.haigh-2020@hull.ac.uk (T.H.); h.beattie-2018@hull.ac.uk (H.B.); m.wade@hull.ac.uk (M.A.W.); j.greenman@hull.ac.uk (J.G.); 2Department of Otorhinolaryngology, Head and Neck Surgery, Hull University Teaching Hospitals NHS Trust Hull, Hull HU16 5JQ, UK; james.england3@nhs.net; 3School of Engineering, University of Hull, Cottingham Rd., Hull HU6 7RX, UK; d.kuvshinov@hull.ac.uk; 4Department of Pathology, Hull University Teaching Hospitals NHS Trust Hull, Hull HU3 2JZ, UK; laszlo.karsai@nhs.net

**Keywords:** miRNA, thyroid, Graves’, PTC, extracellular vesicles, tissue on chip

## Abstract

Small extracellular vesicles (sEVs) contain microRNAs (miRNAs) which have potential to act as disease-specific biomarkers. The current study uses an established method to maintain human thyroid tissue ex vivo on a tissue-on-chip device, allowing the collection, isolation and interrogation of the sEVs released directly from thyroid tissue. sEVs were analysed for differences in miRNA levels released from benign thyroid tissue, Graves’ disease tissue and papillary thyroid cancer (PTC), using miRNA sequencing and quantitative reverse transcriptase polymerase chain reaction (qRT-PCR) to identify potential biomarkers of disease. Thyroid biopsies from patients with benign tissue *(n* = 5), Graves’ disease (*n* = 5) and PTC (*n* = 5) were perfused with medium containing sEV-depleted serum for 6 days on the tissue-on-chip device. During incubation, the effluents were collected and ultracentrifuged to isolate sEVs; miRNA was extracted and sequenced (miRNASeq). Out of the 15 samples, 14 passed the quality control and miRNASeq analysis detected significantly higher expression of miR-375-3p, miR-7-5p, miR-382-5p and miR-127-3p in the sEVs isolated from Graves’ tissue compared to those from benign tissue (false discovery rate; FDR *p* < 0.05). Similarly, miR-375-3p and miR-7-5p were also detected at a higher level in the Graves’ tissue sEVs compared to the PTC tissue sEVs (FDR *p* < 0.05). No significant differences were observed between miRNA in sEVs from PTC vs. those from benign tissue. These results were supported by Quantitative Reverse Transcriptase Polymerase Chain Reaction (qRT-PCR). The novel findings demonstrate that the tissue-on-chip technology is a robust method for isolating sEVs directly from the tissue of interest, which has permitted the identification of four miRNAs, with which further investigation could be used as biomarkers or therapeutic targets within thyroid disease.

## 1. Introduction

Early diagnosis of disease is one of the most important factors that leads to improved patient outcome. Tissue sampling is invasive but remains the gold standard for diagnosis within solid organs [1]. Patients with thyroid disorders, such as Graves’ disease and thyroid cancer, would benefit from the identification of highly sensitive, specific biomarkers detectable in biofluids. This would enable the early detection of disease and provide a minimally invasive mechanism for disease monitoring, during and post-treatment, resulting in evidence-based, clinical management.

Thyroid carcinoma is the most common endocrine malignancy, and ranks the 9th most prevalent cancer worldwide, affecting 1–5% of women and ~2% of men [2]. Papillary thyroid cancer (PTC) is the most common subtype of thyroid cancer, constituting up to 90% of thyroid cancers worldwide [3]. PTC is detected through the combination of ultrasound imaging and fine needle aspiration of suspected thyroid nodules. Treatment follows a multidisciplinary team meeting and may involve surgery and/or radioiodine ablation, with measurement of serum thyroglobulin to monitor remnants and recurrence. However, limitations with the sensitivity and specificity of these diagnostic tools and disease monitoring biomarkers are apparent and they may fail to determine differences between either malignant and benign neoplasms, or between different thyroid cancer subtypes [4,5]. With improvements in diagnostic techniques over recent years, there is much controversy concerning the ‘over treatment’ of PTC [6], which could be circumvented with more effective biomarker monitoring [7].

Graves’ disease is an autoimmune disease of the thyroid and the most common cause of hyperthyroidism, occurring in 1–1.5% of the population [8]. Graves’ disease is diagnosed through patient history and clinical examination and confirmed through biochemical tests which detect elevated thyroid hormones and thyroid-stimulating hormone receptor antibodies (TSH-R-Ab) [9]. Although well managed in the majority of cases, using thionamides, surgery or radioiodine therapy, approximately 30% of patients develop Graves’ Orbitopathy (GO) with 6% developing severe GO that has the potential to cause loss of sight [10]. Currently, high TSH-R-Ab levels help identify patients at risk from GO, but additional biomarkers for both Graves’ disease and GO would increase accuracy and help in the development of personalised treatment [11].

Non-coding microRNAs (miRNAs) have the potential to be used as minimally invasive biomarkers, as they are present in bodily fluids, either bound to proteins or encapsulated in extracellular vesicles (EV) [12,13]. miRNAs are key players in intercellular communication both locally and systemically and their primary role is gene regulation, allowing them to control many cellular processes [14,15]. Dysregulation of miRNA can lead to the development of disease, promoting many of the hallmarks of cancer [16,17], making them ideal candidates for biomarkers and therapeutic targets [18]. Exosomes, or small EVs (sEVs), contain miRNA, as well as protein, lipids, other RNA and DNA, and are actively released from cells and possess a molecular profile representative of their cellular source [19]. sEVs are elevated in the blood of patients both with thyroid cancer [20] and Graves’ disease [21].

To date, the only published clinical data have come from studies of patients’ serum/plasma and these have shown the dysregulation of miRNA in PTC-derived sEVs, compared to benign sEVs [22]. The ability to distinguish between follicular thyroid cancer and hyperplastic nodules [23], follicular cancer and PTC [24], as well as between Graves’ disease patients with and without Graves’ orbitopathy and healthy controls [11] by measuring changes in miRNAs in thyroid tissues, serum/plasma and in sEVs has also been demonstrated. Detailed characterisation of thyroid-derived sEVs and their content is necessary to identify an miRNA signature characteristic of specific thyroid disease.

Although serum and plasma are a rich source of EV, they contain a mix of vesicles derived from various cell types of both healthy and diseased origin, making characterisation complicated and those derived from cell lines are from a single cell source that does not represent the multicellular nature of tissue.

The group in Hull have developed a unique tissue-on-chip device for the successful maintenance of thyroid tissue (Figure 1) [25]. The use of a tissue-on-chip device to maintain thyroid biopsies from patients provides the potential for isolating and characterising EVs known to originate only from the thyroid and associated cells in the biopsy.

The aim of the study was to determine whether disease-specific sEVs could be isolated from different thyroid pathologies maintained on the perfusion device and whether distinct EV miRNA signatures exist between the three thyroid pathologies investigated (benign, Graves’ and cancer). Both Graves’ and cancer sEVs have been investigated in relation to benign tissue to understand the overlap in miRNA expression between these disease states.

## 2. Results

### 2.1. Tissue Morphology (Hematoxylin and Eosin; H&E)

As demonstrated previously [25], the Hull-designed precision cut tissue perfusion device maintained the morphology of the thyroid tissue for 6 days, as verified by pathological assessment (Dr L. Karsai) of the H&E stained tissue pre- vs. post-perfusion tissue (Figure 2).

### 2.2. Nanoparticle Tracking Analysis (NTA)

Over the 6-day maintenance period on the tissue-on-chip device, the effluent coming from the chip was collected daily and analysed using NTA for both particle size and concentration. The average size of particles released from the benign, Graves’ and cancer tissue samples, over the 6-day period was 116.9 ± 16.8 nm, 140.1 ± 41.4 nm and 131.3 ± 16.8 nm respectively, with a range between 80 nm and 187 nm, which falls within the expected range for exosomes/sEVs [26]. There was no significant difference in size between pathologies or over time within each pathology (Figure 3a). The concentration of particles between samples was very variable, even when taking into account the different weights of starting material, ranging between 1.1 × 10^6^ and 2.84 × 10^8^ particles/mL/mg. Although the means of the benign sample were greater than the other pathologies on 5 out of 6 days of measurement, the variation in the data, with this small sample size, meant that this was not significant, either between the pathologies or over time on the perfusion device (Figure 3b).

### 2.3. Western Blotting

The effluents from the tissue-on-chip devices were combined from all chips from three patient samples (1 Hürthle cell carcinoma, 1 Graves’ and 1 tumour), from day 2 onwards, resulting in approximately 14 mL per chip, with the sEVs isolated by ultracentrifugation. Day 1 effluent was omitted so that debris from the device setup was excluded. The protein concentration obtained from the sEV lysates ranged from 0.137 to 0.189 µg/µL. With the protein concentrations being low, the maximum volume of 60 µL was added to the sEV wells to maximise detection. The classic EV markers, tetraspanins Cluster of Differentiation (CD) 9, CD63 and CD81, were used to characterise the isolated sEVs [27]. CD63, which can range in size between 30–60 kDa depending on glycosylation, was detected in both the tissue lysate and the sEV lysate (Figure 4a) as was CD9 (24 kDa). In contrast, CD81 was only detectable in the tissue lysate (Figure 4b).

### 2.4. miRNA Sequencing

RNA was extracted from the sEV pellets obtained from the ultracentrifugation of effluent collected from 15 patient samples maintained on the PCTS device (benign *n* = 5, Graves’ *n* = 5 and PTC *n* = 5; Table 1). The RNA was exported to Qiagen (Hilden, Germany) for miRNA sequencing. The Qubit RNA high-sensitivity (HS) assay initially revealed 7 samples with RNA concentrations below the lower limit of detection (LOD) (Table 1). Despite this, all samples underwent qPCR quality control (QC) and 14 out of the 15 samples were deemed to be of satisfactory quality for miRNA sequencing.

Around 31 million reads were obtained per sample during miRNA sequencing and a high median Phred score of 34 was obtained for all samples, demonstrating miRNA of sufficient quality.

Differential gene expression (DEG) analysis was undertaken between the sEVs isolated from the three pathologies to determine significant differences in miRNA expression. Mean Average (MA) plots were generated to visualise differences in expression (Figure 5). Comparison of miRNA expression in sEVs released from Graves’ patient tissue (*n* = 4) with that in sEVs released from benign thyroid tissue (*n* = 5) demonstrated that hsa-miR-375-3p (false discovery rate [FDR], *p* = 0.0004), hsa-miR-7-5p (FDR, *p* = 0.017), hsa-miR-382-5p (FDR, *p* = 0.032) and hsa-miR-127-3p (FDR, *p* = 0.032) were significantly up-regulated in Graves’ sEVs compared to benign sEVs (Figure 5a). Similarly, comparison of miRNA expression in sEVs isolated from Graves’ (*n* = 4) and PTC (*n* = 5), demonstrated that miR-7-5p (FDR, *p* = 0.0031) and miR-375-3p (FDR, *p* = 0.013) were significantly higher in the sEV from Graves’ tissue relative to the sEV isolated from thyroid cancer (Figure 5b). In contrast, no significant changes in miRNA levels were observed between the sEVs isolated from malignant thyroid tissue (*n* = 5) versus the sEVs from benign thyroid tissue (*n* = 5; Figure 5c).

### 2.5. Quantitative Reverse Transcriptase Polymersase Chain Reaction (qRT-PCR)

The four miRNAs which were found to be significantly different between the sEVs isolated from different thyroid pathologies through RNA sequencing (FDR, *p* value < 0.05) were investigated further using qRT-PCR on a set of 17 samples (Benign, *n* = 5; Graves’, *n* = 6; and PTC, *n* = 6). A panel of five stable miRNAs (miR-16-5p, 320c, 191-5p, Let 7e-5p, Let 7c-5p) were identified from the sequencing data and through preliminary testing using qRT-PCR. miR-16-5p and miR-320c were selected as being the most stable to act as reference genes; these had average C_T_ values of 30.9 and 31.3, respectively. Despite attempting to add the same amount of RNA into the Complementarty DNA (cDNA) synthesis mix, some variation in the stable genes was observed, highlighting the need for normalisation. For each of the miRNA of interest (miR-375-3p, miR-7-5p, miR-382-5p and miR-127-3p), the mean C_T_ value was ≥30 for all samples demonstrating relatively low expression of these miRNA in thyroid tissue sEVs. However, following normalisation against the stable genes and analysis through the geNorm software (Qiagen; https://geneglobe.qiagen.com/gb/analyze; accessed in 8 October 2023), using the 2^−∆∆CT^ method, a significant increase in miR-375-3p, miR-382-5p and miR-127-3p was observed in the sEVs derived from PTC tissue, compared to those released from benign tissue (Table 2). An increase in both miR-375-3p and miR-382-5p was also seen in the Graves’-derived sEVs compared to the benign sEVs; however these increases only approached significance (*p* = 0.0788 and *p* = 0.0683, respectively). In contrast, despite having a positive fold change/regulation (Table 2), the miR-7-5p was not significantly different between PTC and benign sEVs or between Graves’ and benign sEVs. No significant changes in the two miRNAs of interest were observed between the sEVs derived from Graves’ and those from PTC tissue using qRT-PCR.

## 3. Discussion

This study describes the successful use of tissue-on-chip technology to maintain thyroid tissue of different pathologies, allowing for the isolation of sEVs originating directly from the tissue. This is the first study to sequence the miRNA content of sEVs released from thyroid tissue using this method and has identified four miRNAs (miR-375-3p, miR-7-5p, miR-127-3p and miR-382-5p) of particular interest between the three thyroid pathologies investigated.

NTA-detected particles of the size equating to sEVs/exosomes (80 nm to 187 nm) [26], and Western blot confirmed the presence of the exosomal markers CD63 and CD9 in the sEV lysate. However, CD81 was only present in the tissue lysate and not the sEV lysate in the three samples analysed. This is in accordance with the previous literature that found the tetraspanins not to be homogeneously or ubiquitously distributed [28] and that the expression of these markers can vary depending on the cell source. For example, natural killer cell-derived sEVs have been found to be devoid of CD9, whereas platelet-derived sEVs were devoid of CD81 [28]. In addition, Mizenko et al. [29] found that only 20% of sEVs isolated from the serum of ovarian cancer patients expressed CD81 compared to CD63 and CD9 which were expressed on 30% and 60%, respectively. Therefore, it is likely that if the expression of CD81 was low on the sEVs isolated from the tissue-on-chip experiments, the sensitivity of Western blotting would not be sufficient to detect it.

The miRNA which showed the most difference from the sequencing results was miR-375-3p which was significantly increased in Graves’ sEVs compared to both benign and cancer-derived sEVs. This up-regulation in expression of miR-375-3p in Graves’ sEVs compared to benign sEVs was also confirmed by qRT-PCR but this only approached significance, perhaps due to the relatively small cohort size. Interestingly, despite there being no significant difference in miRNA-375-3p expression in cancer-derived sEVs compared to benign sEVs in the sequencing analysis, this comparison using qRT-PCR showed an elevated level of miR-375-3p in the cancer sEV. This elevation of miRNA-375-3p in both Graves’ and cancer pathologies of the thyroid is in agreement with the literature, which mainly focusses on medullary thyroid cancer (MTC), where Censi et al. [30] found miR-375-3p at a 101 times higher level in the plasma of patients with MTC (*n* = 68) compared to the plasma of healthy individuals (*n* = 57). This also corroborates the work by Romeo et al. [31] who showed that out of 51 miRNAs found to be elevated in MTC patient tissue compared to controls using microarray, miR-375p was the most overexpressed, and that the levels of miRNA-375 in the plasma of MTC patients (*n* = 36) were again significantly higher than controls (*n* = 36). Similarly, miR-375 was one of four miRNAs (miR-375, miR-22, miR-16 and miR-451) that were significantly elevated in the serum of Graves’ disease patients (*n* = 17) compared to healthy controls (*n* = 20) [32]. Interestingly, miR-16 was one of the miRNAs that was stably expressed across all of the samples in the current miRNA sequencing analysis. The importance of elevated levels of miR-375 in thyroid disease compared to controls is highlighted in the study by Shi et al. [33], who used both the Gene Expression Omnibus and 12 online prediction databases to identify 1132 overlapping, prospective targets for miR-375. Shi et al. [33] also discovered that the most enriched terms found using Gene Ontology analysis were negative regulation of transcription from RNA polymerase II promoter, golgi membrane and pathway of protein binding, with Kyoto Encyclopedia of Genes and Genomes (KEGG) pathway analysis identifying the PI3K/Akt signaling pathway as the most enriched in the miR-375 target genes, suggesting that miR-375 may have a role to play in disease pathogenesis. These data are in contrast with the data found in PTC (similar to those used in this study) where miR-375 was significantly down-regulated compared to adjacent normal tissue (*n* = 60 for each cohort) [34]. Wang et al. [34] also reported that miR-375 can inhibit proliferation, increase apoptosis and reduce migration and invasion in vitro. These opposing findings fit with the literature which states that miR-375 has a multi-functional role in many processes, including immunity, inflammation and cancer [35], and more work is needed to identify and delineate a role for miR-375 within thyroid disease pathogenesis.

MiR-7-5p was also found to be elevated in sEVs derived from Graves’ tissue compared to both benign and cancer-derived sEVs. However, the PCR data did not validate this result and no significant difference was observed between the cancer and the benign samples. The literature, however, states that miR-7-5p is down-regulated in PTC compared with matched normal adjacent tissue [36,37] (*n* = 10 and *n* = 14, respectively). Notably, Saiselet et al. [37] also found miR-375 to be up-regulated in tumour tissue compared to normal tissue. miR-7-5p has been shown to inhibit cell proliferation in vitro [36,38], following transfection into PTC cell lines and there is a negative correlation between miR-7-5p and insulin receptor substrate 2 and epidermal growth factor receptor expression, suggesting a tumour suppressive role for miR-7-5p [39]; however, oncogenic roles for miR-7-5p have also been described [38]. Duan et al. [40] suggest that the down-regulated level of miR-7-5p in formalin-fixed PTC samples from 101 patients compared to 40 nodular goitre control samples, which correlated with the aggressiveness of the tumour, makes it an ideal candidate for diagnostic purposes. Although the current study did not find a difference between cancer and benign tissues, the level of miR-7-5p was lower in sEVs from cancer than Graves’ tissue-derived sEVs. This may be due to the fact that in vivo changes to sEV profiles could be secondary to the effects of the disease, whereas the effluent from the devices detects sEVs released specifically from the tissue biopsy.

The final two miRNAs that showed any significant difference between thyroid pathologies were miR-127-3p and miR-382-5p which were both up-regulated in the sEVs isolated from Graves’ tissue compared to those from benign tissue, according to the sequencing data. This elevation was maintained in the PCR results for miR-382-5p but the difference failed to reach significance. However, similar to the miR-375-3p, both miR-127-3p and miR-382-5p were found to be significantly up-regulated in sEVs isolated from PTC compared to those from benign thyroid sEVs. This was in agreement with the literature for miR-127, which has been found to be up-regulated in both MTC (*n* = 15 MTC vs. adjacent normal; Ref. [41]) and PTC (*n* = 118 PTC vs. adjacent normal; Ref. [42]), where the over-expression correlated with advanced tumour stage and poor prognosis. Sun et al. [42]) also demonstrated that over-expression of miR-127 in thyroid cancer cell lines led to increased cell proliferation, migration and invasion through the Replication Inhibitor 1 protein (REPIN1), which is required for the initiation of chromosomal DNA replication. To the best of the authors’ knowledge, no literature exists describing an association between miR-382 and either thyroid cancer or Graves’ disease tissue. However, it has been shown to be down-regulated in other cancers, including colorectal [43] and hepatocellular carcinoma [44] tissue and over-expression in both colorectal cancer and hepatocellular carcinoma cell lines has demonstrated reduced proliferation, colony forming activity and migratory capacity, pointing towards a tumour suppressive role for this miR [43,44,45]. In breast cancer, miR-382 is significantly down-regulated in tumour-associated macrophages and M2-polarised macrophages from the patient tissue and in vitro, over-expression of miR-382 inhibits M2 polarisation and inhibits the ability of tumour associated macrophages to promote malignant behaviour [46], again suggesting a tumour suppressive role for miR-382. Despite there being studies describing the potential use of some miR as biomarkers for Graves’ disease, including miR-144 (down-regulated) and miR-762 (up-regulated) [45] and miR146a (down-regulated), miR-155 (down-regulated) and miR-210 (up-regulated) [47], there is no literature describing a role for miR-7, miR-127 or miR-382 in Graves’ disease.

In order to determine the usefulness of miR-375, miR-7, miR-382 and miR-127 as biomarkers in thyroid disorders, further comparisons of the levels of the four miRNAs is required in the serum of healthy individuals compared to both Graves’ patients and cancer patients with different thyroid cancer subtypes and at different tumour stage, to determine whether they can differentiate not only between pathologies but also between stages of disease in vivo.

In conclusion, the use of tissue-on-chip technology allows the search for miRNA biomarkers in thyroid disease to be more focused specifically on the tissue of interest and has identified four candidates to be investigated further. It is hypothesized that treatment will modify levels of sEV miRNA, which will provide an indication of treatment response and aid with the monitoring of disease in terms of cure or recurrence.

## 4. Materials and Methods

Ethical approval for the study was obtained from the National Research Ethics Service, North East Newcastle and North Tyneside (15/NE/0412) and from Hull University Teaching Hospitals NHS Trust Research and Development (R1925). Tissue samples were obtained from patients undergoing thyroid surgery and thyroid function test, TRAb analysis, thyroid ultrasound scans and fine needle aspiration cytology results were reviewed, which provided clarification of the thyroid pathology being resected; all patients were treatment naïve. A total of 29 patients were included in the study, 19 females and 10 males, with an age range of 19–83 years (Table 3).

### 4.1. Tissue Processing and Preparation

Upon resection, the thyroid sample was placed in Dulbecco’s Modified Eagle Medium (DMEM; GE Health-care, Yeovil, Somerset, UK) containing 10% (*v*/*v*) foetal bovine serum (FBS; Labtech International, Heathfield, East Sussex, UK), penicillin/streptomycin (100 U/mL and 100 mg/mL respectively; Corning, Flintshire, UK) and 0.4 mM glutamine (GE healthcare) and immediately transferred to the on-site laboratory.

A sample of the tissue was placed in 4% (*w*/*v*) paraformaldehyde (PFA, Merck/Sigma, Dorset, UK) for tissue fixation. The remainder of the tissue was immobilised onto a tissue holder using superglue and sliced at a thickness of 350 μm in ice-cold phosphate-buffered saline (PBS) containing Amphotericin B (Scientific Laboratory Supplies [SLS], Nottingham, UK) and penicillin/streptomycin (100 IU/mL and 100 µg/mL respectively, Corning), using a vibratome (Leica VT1200S, Milton Keynes, UK) with a blade speed of 0.1 mm/s and amplitude of 2.5 mm. A skin biopsy punch (Stiefel, Middlesex, UK) was used to generate precision cut tissue slices (PCTS) of 5 mm in diameter. Each of the PCTS were weighed individually prior to insertion into their respective tissue-on-chip device (Figure 1). Prior to each experiment, the device was flushed with 70% ethanol and PBS.

Complete DMEM, as described above, substituted with 10% (*v*/*v*) exosome depleted FBS (Labtech International, East Sussex, UK) was loaded into a 20 mL syringe (Becton Dickinson, Wokingham, UK) and connected, using a Female Luer-Lock™ adapter (Mengel Engineering), to the ETFE tubing via a 0.22 μm filter (Sarstedt, Leicester, UK; Figure 2).

### 4.2. Setting up and Running Perfusion Devices

The prepared PCTS were loaded onto a 70 μm porous nylon membrane (FALCON, Corning Brand, Durham, UK) positioned on top of the sintered disc within the inlet PEEK plate. The outlet PEEK plate was then secured in place using metal screws (RS components, Leeds, UK). The syringe was connected to a Harvard PhD 2000 syringe pump (Harvard, Cambridge, UK), which provided a pressure-driven perfusion rate of 2μL/min. The tissue-on-chip device was maintained at 37 °C, under constant perfusion for 6 days (144 h). Medium coming off each chip (effluent) was collected in 15 mL polypropylene tubes (Sarstedt, Leicester, UK) on a daily basis. An aliquot (1 mL) of the effluent medium was stored at 4 °C for nanoparticle tracking analysis and the remainder stored at −80 °C prior to ultracentrifugation. The remaining DMEM within the 20 mL input syringe at day 6 was kept for subsequent analysis to control for bovine extracellular vesicles.

Fresh and post-perfusion thyroid tissue samples were fixed in 4% (*w*/*v*) PFA for 24 h. PFA-fixed tissues were dehydrated and embedded in molten paraffin wax (Epredia™ Histoplast Paraffin PE, Fisher Scientific, Loughborough, UK) and prepared for H&E staining as described previously [48]. Tissue morphology both pre and post chip was examined for integrity by a head and neck pathologist (Dr L Karsai).

### 4.3. Nanoparticle Tracking Analysis

The size and concentration of EVs in both the effluent (1 mL) collected daily from the tissue-on-chip system and in the syringe medium on day 6 were determined using an LM10 Nanosight NTA System (Malvern Panalytical, Malvern, UK) fitted NTA software ( v.3.4; open source; https://www.malvernpanalytical.com/en/support/product-support/software/nanosight-nta-software-update-v3-2; accessed on 15 February 2022). Three captures of 60 s each were performed and mean size (nm) and particles/mL were recorded. The samples were diluted 1:8 in serum-free medium to ensure the concentration was within the range detectable by the machine. Significant differences between pathologies and over time were assessed using two-way ANOVA with Bonferroni post-hoc correction (GraphPad/Prism 9; https://www.graphpad.com; accessed on 21 March 2022).

### 4.4. Isolation of Total Extracellular Vesicles

sEVs were isolated from the tissue effluent coming from the perfusion device using sequential centrifugation steps [27]. The effluent from day 2 to day 6 was combined from all devices (day 1 was excluded as this contained particles released during processing), filtered through a 0.2 µm filter and centrifuged at 400× *g* for 10 min at 4 °C to remove cell debris (Eppendorf 5810R, Stevenage, UK). The supernatant was then subjected to further sequential centrifugation steps (2000× *g* 10 min and 10,000× *g* at 4 °C for 30 min; Beckman Coulter Ltd., High Wycombe, UK), before transfer into OptiSeal centrifuge tubes (Beckman Coulter) for ultracentrifugation. A TLA-110 fixed angle rotor was used at 100,000× *g* at 4 °C for 1 h (Beckman Optima MAX-XP). The sEV pellet was washed with ultrafiltered (20 nm) Phosphate Buffered Saline (PBS) and ultracentrifuged again for 1 h to generate an sEV pellet. The final sEV pellet was briefly air dried by inversion before storage at −80 °C prior to protein or RNA extraction.

### 4.5. Protein Extraction and Western Blotting for EV Markers

Protein was extracted from sEVs in the pellets generated from ultracentrifugation using 100 μL of ice-cold Radioimmunoprecipitation assay buffer (RIPA; SLS) containing phosphatase inhibitor cocktail (PhosSTOP™), and protease inhibitor cocktail (cOmplete™ ULTRA) tablets (both Merck/Sigma, Dorset, UK). Pellets from replicate tubes (from day 2 onwards) were lysed in a total of 100 µL of RIPA buffer to concentrate the protein. Lysates were incubated for 15 min on ice before vortexing and sonication for 3 min. Debris was removed by centrifugation at 4 °C for 15 min at 16,000× g and the supernatant was analysed for protein content using the PierceTM™ BCA protein assay kit following the manufacturer’s instructions (Thermofisher Scientific, Loughborough, UK). Tissue lysates were prepared in the same way but using 300 µL of RIPA buffer and a tube pestle to help lyse the cells.

A non-reducing sodium dodecyl sulphate (SDS), polyacrylaminde gel electrophoresis method [49], using the Bolt™ System (Thermofisher) was used to detect the classic EV markers CD9, CD63 and CD81.

Where possible, 5 µg of protein was combined with an equal volume of 2 × non-reducing Bolt™ sample buffer. For those samples with low-protein concentrations, the maximum volume of 60 µL was used and 10 µL of 4 × Bolt™ sample buffer was added. The lysate/sample buffer mix was heated at 70 °C for 10 min and loaded onto a Bolt™ 4–12% Bis/Tris gel (Thermofisher) alongside both Sea Blue (6 µL) and Magic Mark (3 µL) protein ladders (Thermofisher). Electrophoresis was achieved in 1 × MES SDS running buffer (provided) for approximately 40 min at 150 V until the dye front approached the end of the gel.

The gel was transferred to a Polyvinylidene fluoride (PVDF) membrane (Biorad, Watford, UK) and the Transblot^®^ Turbo™ semi-dry transfer system was used to transfer proteins at 25 V for 30 min. Following the transfer, the membrane was blocked for 1 h in 5% (*w*/*v*) milk powder (SLS) in PBS-Tween-20 0.1% (*v*/*v*) (Merck/Sigma) at 4 °C with end-to-end rocking. Primary antibodies (5 mL), CD9 (Mouse IgG1 anti-human CD9, clone Ts9; Invitrogen, Paisley, UK), CD63 (Mouse IgG1 anti-human CD63, clone Mx-49.1295, Insight Biotechnology, Wembley, UK) and CD81 (Mouse IgG2a anti-human CD81, clone B-11; Insight Biotechnology) were added to the membrane (1:500) in blocking buffer and incubated overnight at 4 °C with end-to-end rocking.

The membrane was washed (3 × 10 min) in PBS-Tween-20 0.1% (*v*/*v*), before the addition of IgGκ binding protein linked to Horse Radish Peroxidase (m-IgGκ BP-HRP; 1:5000; In-sight Biotechnology) in 5% (*w*/*v*) milk powder for 1 h at room temperature with gentle rocking. Following further washes, bands were visualised using SuperSignal™ West Pico PLUS Chemiluminescent Substrate (Thermofisher; prepared by adding 1 mL of Regent A to 1 mL of Reagent B) and autoradiography with exposure for 5 min. The X-ray film was developed using Ilfosol 3 developer and fixer (Ilford, Mobberley, UK).

### 4.6. Extraction of RNA from sEV Using the QIAGEN miRNeasy Microkit

RNA was extracted from the sEV pellets using the QIAGEN miRNeasy microkit (QIAGEN, Hilden, Germany), following the manufacturer’s instructions. Qiazol lysis reagent (Qiagen; 700 µL) was added to one of the sEV pellets, the tube was vortexed for 2 × 30 s before the lysate was transferred to replicate tubes and the process repeated, in order to concentrate the RNA. Phenol: chloroform extraction was performed as directed, with precipitation of the upper RNA containing aqueous phase using 100% ethanol. RNA was purified using RNeasy MinElute spin columns before elution in 14 µL of RNase-free water to generate 12 µL of eluate. RNA was quantified in 1 µL using the Biochrom SimpliNano™ Spectrophotometer (VWR, Lutterworth, UK).

### 4.7. miRNA Sequencing

Prepared RNA was shipped to Qiagen (Hilden, Germany) and the QIAseq miRNA Library Kit (Qiagen) was used to convert 1 ng or 5 µL (where the concentration was low) of total RNA into miRNA NGS libraries. qPCR was used to quantify the library pools and sequenced on a NextSeq (Illumina Inc., San Diego, CA, USA) sequencer. Raw data were de-multiplexed and FASTQ files for each sample were generated using the bcl2fastq2 software v 2.20 (Illumina Inc.). All primary analyses were carried out using CLC Genomics Server 22.0.2 and reads were mapped to miRBase version 22. The ‘Empirical analysis of DGE’ algorithm was used for differential expression analysis with default settings (Qiagen).

For all unsupervised analysis, only miRNAs were considered with at least 10 counts summed over all samples. MA plots were generated to visualise differences between the three thyroid pathologies, by transforming the data into M (log_2_ ratio) and A (log_2_ mean average) scales, before plotting these values against each other.

### 4.8. qRT-PCR Using Individual miRCURY LNA (Locked Nucleic Acid) PCR Assays

Results from the RNA sequencing were validated using qRT-PCR using individual SYBR green miRCURY LNA PCR assays (Qiagen). cDNA was first synthesised from sEV RNA extracted from 17 different patient tissue-on-chip samples (Table 3) using the miRCURY LNA RT Kit, following the manufacturer’s instructions (Qiagen) with the UniSp6 spike in (1 µL/20 µL reaction mix) as a control for reverse transcription efficiency. The reaction mix was incubated for 60 min at 42 °C with a 5 min denaturation step at 95 °C. The miRCURY LNA RT Kit allows polyadenylation of the miRNA and reverse transcription in a single step. For each sample, 50 ng of RNA was added for cDNA synthesis and a no-template control was included to control for contamination. qRT-PCR was carried out as described previously [50], using LNA-optimised, SYBR^®^ Green-based miRNA PCR primers (Qiagen) specific for the miRNA which showed significant differences (*p* < 0.05), between thyroid pathologies in the miRNA sequencing (miR-375-3p, miR-7-5p, miR-382-5p and miR-127-3p). Two stable miRNAs, identified through the sequencing work (miR-16-5p and miR-320c), were also included in each PCR run as reference miRNA. The prepared cDNA was used at a 1:30 dilution and the PCR was prepared in 96-well plates (Applied biosystems/Fisher Scientific MicroAmp Fast Optical 96-well reaction plate). The PCR was carried out on an ABI StepOnePlus™ system (Applied Biosystems, Warrington, UK) with an initial incubation step of 2 min at 95 °C, with 40 cycles of two-step cycling; 95 °C for 10 s and 56 °C for 60 s and a melt curve analysis at 60–95 °C. Initial data analysis used the software supplied to obtain raw C_T_ values. Comparison between the different thyroid pathologies was achieved by uploading the excel file of C_T_ values into the GeneGlobe analysis tool (Qiagen; https://geneglobe.qiagen.com/gb/analyze; accessed in 8 October 2023). Samples were assigned to control (Benign) or test a group (PTC, Graves’) and C_T_ values were normalised based on the geNorm method which uses pre-defined reference miRNAs (miR-16-5p and miR-320c). The analysis tool calculates fold change/regulation using the 2^−∆∆CT^ method to determine differences between groups. To find differences between Graves’ and PTC sEV, Graves’ was assigned as the control group and PTC was the test.

## Figures and Tables

**Figure 1 ijms-25-00071-f001:**
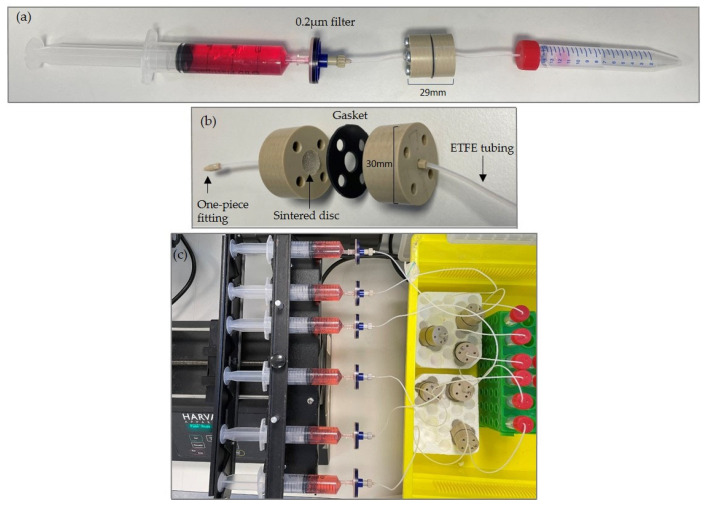
Precision cut tissue slice (PCTS) perfusion device constructed from PolyEther Ether Ketone plastic (PEEK; Direct Plastics, Sheffield, UK). Each plate is 30 mm × 14 mm in size, with threaded axial holes drilled in the centre of each, to accommodate coned adaptors (LabSmith, Mengel Engineering, Denmark), to hold in place both inlet and outlet, ethylene tetrafluoro-ethylene tubing (ETFE; 0.8 mm internal diameter; Kinesis, IDEX Health & Science, Cambridge, UK). Further 1⁄4 inch (6.35 mm) holes were drilled in each PEEK plate so that screws could be inserted to clamp the plates together after sample insertion. A porous sintered Pyrex disc (The Lab Ware-house, Grays, UK) was located in a central cylindrical recess (10 mm × 4 mm) and a silicone gasket (30 mm diameter, 1 mm thickness sheet silicone) with a 6 mm central hole to create a tissue well, was placed between the two PEEK plates. A 70 µm nylon membrane was placed on the glass sintered disc to prevent adherence. (**a**) Complete syringe > PCTS device > collection tube set-up, (**b**) deconstructed PCTS device, (**c**) perfused devices connected the syringe pump and 37 °C incubator.

**Figure 2 ijms-25-00071-f002:**
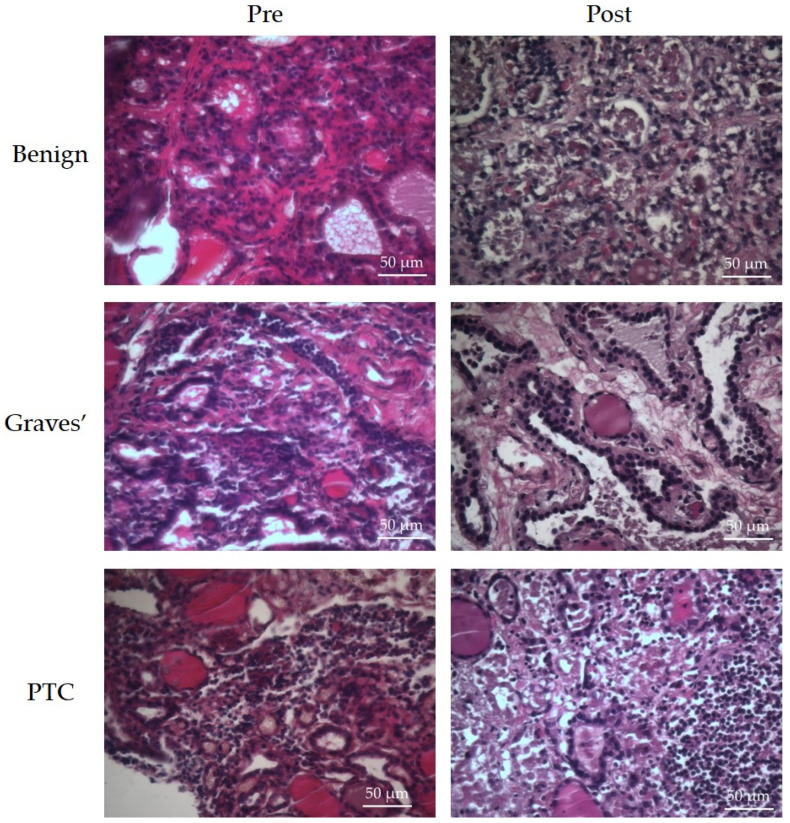
Representative images of formalin-fixed, paraffin embedded benign thyroid, Graves’ and PTC tissue (5 µm thickness) prior to (pre) and following, on-chip culture (post), stained with H&E. ×400 magnification.

**Figure 3 ijms-25-00071-f003:**
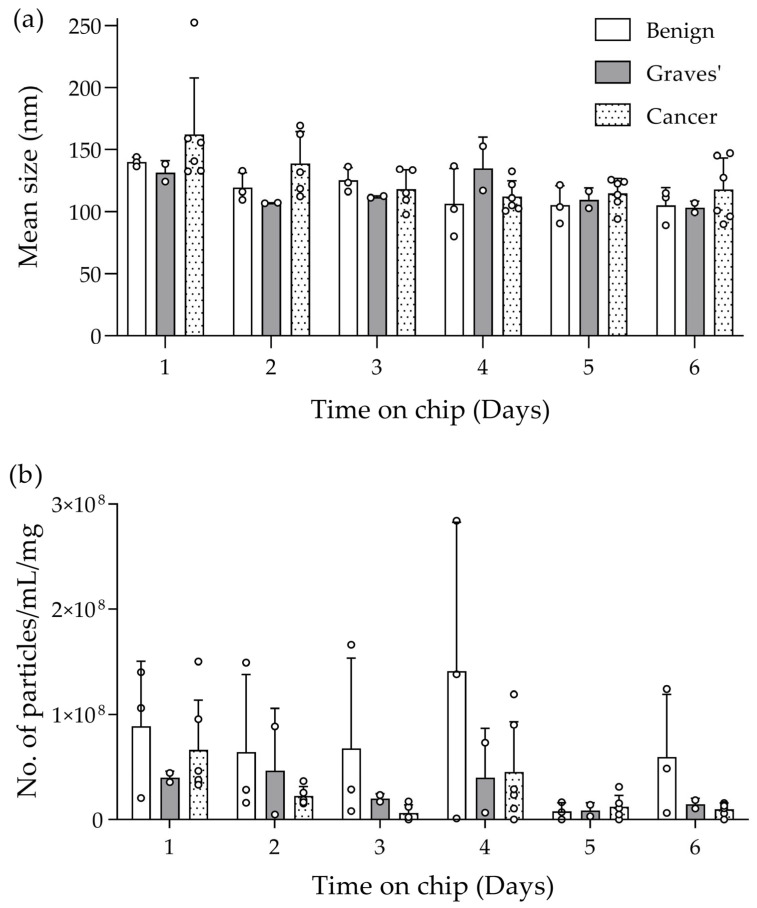
NTA of EVs showing both the size (**a**) and concentration (**b**) of the particles in the effluent (1:8 dilution with serum-free medium) collected from benign (*n* = 3), Graves’ disease (*n* = 2) and thyroid cancer (*n* = 6), tissue maintained for 6 days on a tissue-on-chip device. Mean size (nm) + SD. Mean concentration is expressed as particles/mL/mg starting weight of tissue + SD. Individual data points shown as a circle °. No significant difference in particle size or concentration was observed over time or between tissue types (two-way Analysis of Variance [ANOVA] with Bonferroni post-hoc correction).

**Figure 4 ijms-25-00071-f004:**
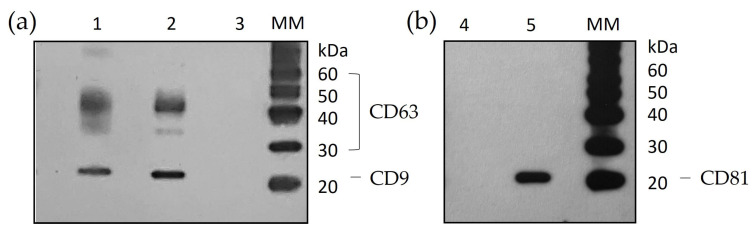
Western blot autoradiographs following 5 min exposure, showing a representative example of (**a**) CD63 (30–60 kDa) and CD9 (24 kDa), (**b**) CD81 (22 kDa) detection. Lanes 1 and 4 sEV lysate, Lanes 2 and 5 tissue lysate, Lane 3 lysate prepared from sEVs isolated from Dulbecco’s Modified Eagle Medium (DMEM) containing only exosome-depleted serum. MM—Magic Mark™ XP Western protein standard. CD63 and CD9 detected in both the whole tissue lysate (Lane 2) and the sEV lysate (Lane 1) obtained from the ultracentrifugation of effluent collected from tissue maintained on the PCTS device. CD81 detected only in tissue lysate (Lane 5).

**Figure 5 ijms-25-00071-f005:**
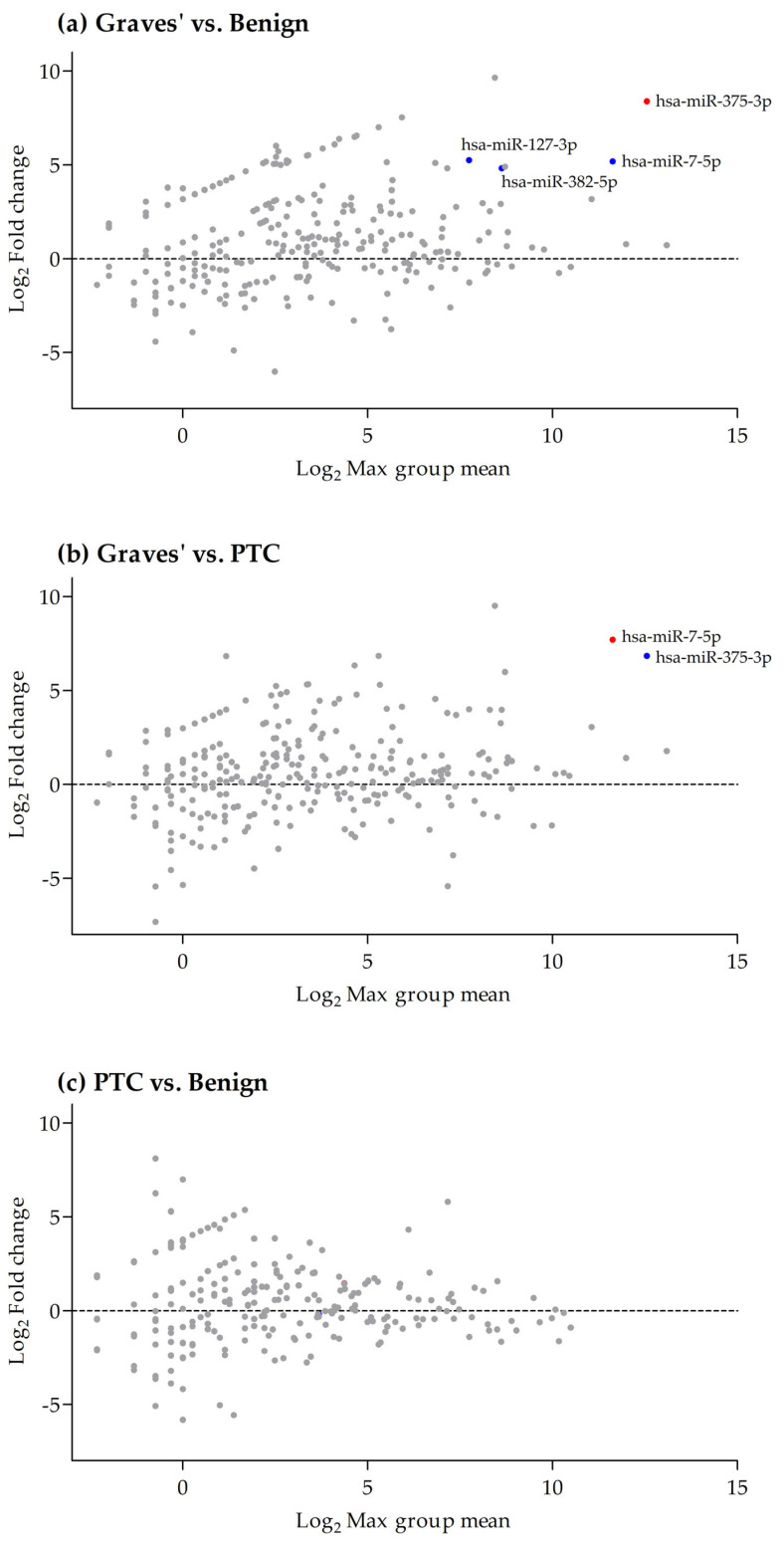
M (Log_2_ ratio) vs. A (Log_2_ Mean average) plots showing the differential expression of miRNA between sEV released from (**a**) Graves’ vs. benign tissue, (**b**) Graves’ vs. PTC tissue and (**c**) PTC vs. Benign tissue. miRNA represented by red dots had a false discovery rate (FDR) *p* < 0.01 and blue dots represent those with an FDR *p* < 0.05.

**Table 1 ijms-25-00071-t001:** RNA concentration of samples prior to miRNA sequencing.

Sample	Sample	RNA Concentration (ng/µL)
2	Graves’	17.5
3 *	PTC	<1.35 *
4 *	PTC	<1.35 *
6 **	Graves’	<1.35 **
7	Benign	11.4
8	PTC	4.6
9 *	Benign	<1.35 *
12 *	Benign	<1.35 *
14	PTC	11
15	PTC	6.9
16	Graves’	4.6
17	Graves’	5.8
18 *	Benign	<1.35 *
19 *	Graves’	<1.35 *
21	Benign	1.8

* Denotes samples with RNA concentration below the lower limit of detection. ** Denotes the sample that did not pass the qRT-PCR QC.

**Table 2 ijms-25-00071-t002:** Results of the geNorm qRT-PCR analysis.

	miRNA	Fold Regulation	*p*-Value
Graves’ vs. benign	miR-375-3p	6.18	0.0788
	miR-7-5p	2.64	0.1235
	miR-382-5p	6.03	0.0683
	miR-127-3p	2.48	0.1371
PTC vs. benign	miR-375-3p	9.90	0.0037 *
	miR-7-5p	2.94	0.4055
	miR-382-5p	6.64	0.0013 *
	miR-127-3p	4.79	0.0189 *
Graves’ vs. PTC	miR-375-3p	0.62	0.1603
	miR-7-5p	0.90	0.2731
	miR-382-5p	0.91	0.1371
	miR-127-3p	0.52	0.1966

* Significant using miRNA 320c and miRNA 16-5p as stable reference miRNA.

**Table 3 ijms-25-00071-t003:** Patient characteristics.

Sample	Tissue Type	Tumour Stage	Age	Gender	Used in
1	Hurthle cell carcinoma	pT2 RO	60	F	WB
2	Graves’	-	60	F	miRNA Seq, qRT-PCR
3	PTC	pT2 N1a	60	M	NTA, miRNA Seq, WB
4	PTC	pT3a N1b R1	83	M	NTA, miRNA Seq
5	Hurthle cell carcinoma	pT2 R0	76	M	NTA
6	Graves’	-	49	F	NTA, miRNA Seq
7	Benign	-	59	F	NTA, miRNA Seq, qRT-PCR
8	PTC	pT2 RO	50	F	NTA, miRNA Seq, qRT-PCR
9	Benign	-	51	F	NTA, miRNA Seq
10	Metastatic FTC	T2N0MO	54	F	NTA
11	PTC	pT3a N1a R1	64	M	NTA
12	Benign	-	75	F	NTA, miRNA Seq
13	Graves’	-	42	M	NTA, WB
14	PTC	pT3B N1a R1	53	M	miRNA Seq, qRT-PCR
15	PTC	pT3a N1b R1	32	F	miRNA Seq, qRT-PCR
16	Graves’	-	40	M	miRNA Seq, qRT-PCR
17	Graves’	-	35	F	miRNA Seq, qRT-PCR
18	Benign	-	55	F	miRNA Seq
19	Graves’	-	70	F	miRNA Seq
20	Benign	-	42	F	qRT-PCR
21	Benign	-	68	F	miRNA Seq, qRT-PCR
22	Benign	-	48	M	qRT-PCR
23	Benign	-	50	F	qRT-PCR
24	Graves’	-	51	F	qRT-PCR
25	PTC	pT3 N1a R2	20	F	qRT-PCR
26	Graves’	-	57	M	qRT-PCR
27	PTC	pT3 N1b	27	M	qRT-PCR
28	PTC	pT3b pN1b R2	19	F	qRT-PCR
29	Graves’	-	51	F	qRT-PCR

M = male, F = female, WB = Western blotting, NTA = nanoparticle tracking analysis, miRNA Seq = miRNA sequencing, qRT-PCR = quantitative reverse transcriptase polymerase chain reaction, PTC = papillary thyroid carcinoma, FTC = follicular thyroid carcinoma.

## Data Availability

Research data are available upon reasonable request to the principal investigators following the University of Hull’s policy.

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
