# Peer review of "The Use of Tissue-on-Chip Technology to Focus the Search for Extracellular Vesicle miRNA Biomarkers in Thyroid Disease"

_ijms, 2023, doi:10.3390/ijms25010071_

Round 1

Reviewer 1 Report

Comments and Suggestions for Authors

Manuscript by Haigh et al. “The use of Tissue-on-chip Technology to Focus the Search for Extracellular Vesicle miRNA Biomarkers in Thyroid Disease” deals with the highly-important point, namely, the search for new EV-enclosed miRNA biomarkers of thyroid tissue disorders, such as  Graves’ disease and papillary thyroid cancer. For the perfusion of thyroid tissue samples and further EV collection, authors used a recently developed set-up, called “a tissue-on-chip device”.

In the work, authors applied a wide set of relevant methods including NTA, Western blotting, miRNA sequencing, miRNA qRT-PCR. Research is well-designed and the results on the differently-expressed miRNAs are quite clear and conclusive.

I have only one minor remark to authors. When showing Western blot data, you clearly demonstrate that CD81 is detected only in tissue lysate and is not detectable in EV lysate (Fig. 4b). Why? I suppose, this is rather unexpected result, because above you write that CD81 (as well as CD9 and CD63) is “the classic EV marker”. So, if you decided to include Fig. 4b in the manuscript, this unexpected result should be explained and interpreted somehow. In the current version, without any comments, it looks a bit bizarre.     

Author Response

Many thanks for your review. Please see all responses in the attached document.

Many thanks

Vicky

Reviewer 2 Report

Comments and Suggestions for Authors

Authors developed a tissue-on-chip to assess sEVs secreted from Graves’ disease and PTC. MiRNA-sequencing and RT-qPCR were employed identify and validate differentially expressed miRNA in tumors compared to benign tissues. Though the overall interpretations are clear, the manuscript will benefit from some changes in statistics and methodologies.

Main points:

  • #248-, comparisons to previous literature is quite interesting and relevant to the purpose of this manuscript. In the current form, it reads like authors are picking specific miRNAs that fits to previous works. Authors should compare all  DEGs of this work and previous literatures to present overlaps e.g. Venn diagrams.
  • Related to this, difference of cellular miRNA profiles in donor cells would be interesting point to assess the validity/similarity of tissue-on-chip model to patient specimens. Selective loading of miRNAs from cells to EV can be also assessed.
  • CD81 is specifically detected only from tissue lysate (Fig 4). Given that CD81 is one of “classic EV markers” (#161), what would be interpretation? Are specific EV population (CD81+) are missing only in the PCTS device, or CD81 expression is missing in actual patient samples as well?

Minor points:

  • Fig. 3. Graves’ disease samples is N=2, which is too few to show SD error bar or statistical text. Particularly with the high variation in the panel B, all raw data points should be shown and power analysis should be employed to test the validity of statistical test. For time series, regression analysis might be more apt than ANOVA.
  • Line#139, “appearing to have” is not objective enough. Authors should perform statistical test or show the actual data in paired way (rather than bar charts) to highlight the point.
  • Line#214. Ct Values does not necessarily convey the expression level since it is affected by primer design differences. Minimally, authors should mention Ct values of other more abundant miRNA in the same context with primers with comparable amplification efficiency + sensitivity. Alternatively, absolute quantification, not ddCT, will be needed to claim the relative difference in abundance.

Author Response

Many thanks for your review. Please see all responses in the attached document.

Kind regards

Vicky Green
